# The Sulfide-Responsive SqrR/BigR Homologous Regulator YgaV of *Escherichia coli* Controls Expression of Anaerobic Respiratory Genes and Antibiotic Tolerance

**DOI:** 10.3390/antiox11122359

**Published:** 2022-11-28

**Authors:** Rajalakshmi Balasubramanian, Koichi Hori, Takayuki Shimizu, Shingo Kasamatsu, Kae Okamura, Kan Tanaka, Hideshi Ihara, Shinji Masuda

**Affiliations:** 1Department of Life Science and Technology, Tokyo Institute of Technology, Yokohama 226-8501, Japan; 2Graduate School of Arts and Sciences, The University of Tokyo, Tokyo 153-8902, Japan; 3Department of Biological Chemistry, Graduate School of Science, Osaka Metropolitan University, Osaka 599-8531, Japan; 4Laboratory for Chemistry and Life Science, Institute of Innovative Research, Tokyo Institute of Technology, Yokohama 226-8501, Japan

**Keywords:** antibiotics, *Escherichia coli*, hydrogen sulfide, reactive sulfur species, SqrR, YgaV

## Abstract

Compositions and activities of bacterial flora in the gastrointestinal tract significantly influence the metabolism, health, and disease of host humans and animals. These enteric bacteria can switch between aerobic and anaerobic growth if oxygen tension becomes limited. Interestingly, the switching mechanism is important for preventing reactive oxygen species (ROS) production and antibiotic tolerance. Studies have also shown that intracellular and extracellular sulfide molecules are involved in this switching control, although the mechanism is not fully clarified. Here, we found that YgaV, a sulfide-responsive transcription factor SqrR/BigR homolog, responded to sulfide compounds in vivo and in vitro to control anaerobic respiratory gene expression. YgaV also responded to H_2_O_2_ scavenging in the enteric bacterium *Escherichia coli*. Although the wild-type (WT) showed increased antibiotic tolerance under H_2_S-atmospheric conditions, the *ygaV* mutant did not show such a phenotype. Additionally, antibiotic sensitivity was higher in the mutant than in the WT of both types in the presence and absence of exogenous H_2_S. These results, therefore, indicated that YgaV-dependent transcriptional regulation was responsible for maintaining redox homeostasis, ROS scavenging, and antibiotic tolerance.

## 1. Introduction

Bacterial flora is microbes whose habitats and compositions significantly influence the health and disease of host humans and animals [1,2]. The enteric bacterium *Escherichia coli* thrives in the gastrointestinal tract of humans and other warm-blooded animals such as pathogenic and commensal bacteria. In such a gastrointestinal environment, oxygen is always a limiting factor for generating energy by respiration. Therefore, under oxygen-limiting conditions, microorganisms derive energy from anaerobic respiration through alternative electron acceptors, such as fumarate, nitrate, and nitrite. Transitions between aerobic and anaerobic growth modes also occur frequently in *E. coli* to facilitate survival in the gastrointestinal environment of host animals. Global alterations in gene expression accompany these metabolic changes, which are controlled by several redox-responsive transcriptional regulators, such as the transcriptional regulator FNR, in addition to a two-component system ArcAB in *E. coli*. These regulators sense the intracellular redox status of cells, in addition to the cell’s extracellular oxygen tension, thereby controlling the expression of over 100 pathway-related genes involved in carbon and electron flow during aerobic and anaerobic transition processes and vice versa [3,4].

Recently, it was established that the redox balance is influenced by gases, such as hydrogen sulfide (H_2_S) and nitric oxide (NO) [5,6,7,8,9,10,11]. In natural environments, the microbiota of many sulfur-oxidizing and sulfate-reducing bacterial species modulate extracellular H_2_S compounds in the biogeochemical sulfur cycle on Earth [12]. Likewise, in the gut microbiota, over 100 bacterial species metabolize sulfur components in host cells. As a result, extracellular H_2_S concentrations in the gut are from ~0.2 to 2.4 mM [7,13]. Intracellular H_2_S concentration is also controlled via several metabolic pathways. For example, 3-mercaptopyruvate sulfurtransferase and cysteine desulfurase are responsible for the bulk of endogenous H_2_S production in *E. coli*, which was shown to be important for the sequestration of free iron necessary to prevent the Fenton reaction, and for protection against the effects of ROS-mediated antibiotic stress [14,15]. H_2_S also behaves as a potent inhibitor of cytochrome oxidase and other metalloproteins [15,16,17,18]. Specifically, sulfide inhibited the activity of the Cu-containing energy-efficient cytochrome *bo* terminal oxidase in *E. coli* [19], which causes ROS production. Under such sulfide-stress conditions, *E. coli* upregulates the expression of another less-energy-efficient cytochrome *bd* oxidase whose activity was not inhibited through sulfides [17,19]. The alteration of the terminal oxidases from *bo* type to *bd* type thus inhibited ROS production [17,19,20,21]. Reactive sulfur species (RSS), such as H_2_S or polysulfanes, effectively inactivates *β*-lactam antibiotics in bacteria [22]. Hence, sulfide contributes to ROS scavenging and redox balance to avoid cell damage especially caused by antibiotics. Extracellular H_2_S was also shown to stimulate respiration and reprogram the central metabolism of the human pathogenic bacterium *Mycobacterium tuberculosis* that leads to enhanced growth, thereby reducing host survival [23]. Thus, these bacteria are proposed to have evolved sensory systems to recognize endogenous and exogenous H_2_S, thereby aiding their ability to regulate cellular functions and metabolism. However, the mechanism is not yet clarified in most bacteria.

We previously identified a sulfide-responsive transcriptional repressor, sulfide-quinone oxidoreductase repressor (SqrR), from the purple photosynthetic bacterium *Rhodobacter capsulatus* [24]. *R. capsulatus* SqrR (*Rc*SqrR) senses extracellular H_2_S and controls genes involved in sulfide-dependent photosynthesis. The *R. capsulatus* mutant, expressing the dominantly positive *Rc*SqrR variant, which constitutively represses the transcription of target genes involved in sulfide oxidation, cannot grow photoautotrophically with H_2_S used as an electron donor [24], indicating the critical role of *Rc*SqrR in photosynthesis regulation. In contrast, SqrR is a homolog of the biofilm growth-associated repressor (BigR), identified in the plant pathogenic bacteria, *Xylella fastidiosa* and *Agrobacterium tumefacience* [25,26]. *X. fastidiosa* BigR (*Xf*BigR) controls the transcription of genes involved in the detoxification of H_2_S derived from associated host plants that are required for adherence to plant roots and biofilm growth [27]. The SqrR/BigR homologs are widely conserved among different bacterial phyla, including *Firmicutes*, *Cyanobacteria*, *Bacteroides*, *Proteobacteria,* and *Actinobacteria* [28]. These results propose that SqrR/BigR homologs control various bacterial physiologies in different types of bacteria, although the physiological significance of their functions is unknown.

Sulfide-sensing mechanisms of SqrR/BigR homologs are also elusive. Hence, *Rc*SqrR senses RSS, such as glutathione-persulfide (GSSH), after which purified *Rc*SqrR makes an intramolecular tetrasulfide bond in the presence of GSSH to negate DNA-binding activities [24]. A similar regulatory mechanism has been also demonstrated for the SqrR homolog of the opportunistic human pathogenic bacterium *Acinetobacter baumannii*, which selectively reacts with GSSH and mainly forms an intramolecular pentasulfide bond [29] for biofilm regulation [30]. Nevertheless, *Xf*BigR can sense sulfides directly, making a disulfide bond when it is exposed to H_2_S to negate its DNA-binding activity [26,27]. Differences in the local structure or fluctuation, as well as altered nucleophilicity of the intermediates of Cys residues conserved in SqrR/BigR homologs, have thus reflected their different selectivities and sensitivities to H_2_S/RSS [29]. However, this hypothesis remains to be confirmed.

To further understand the physiological importance of the H_2_S-derived signal transduction in bacteria, we hereby characterize the SqrR/BigR homolog of the enterobacterium; *E. coli* YgaV. Subsequently, sulfide treatment in vitro and in vivo attenuated the DNA binding of YgaV to a target operator. Genome-wide transcriptome analysis also indicated that YgaV controlled the transcription of anaerobic respiratory genes involved in fumarate, nitrate and nitrate reduction. These results together indicated that YgaV fine-tuned anaerobic respiratory gene expression and ROS homeostasis in response to H_2_S.

## 2. Materials and Methods

### 2.1. Bacterial Strains and Growth Conditions

The *E. coli* strain BW25113 [*rrnB*, DE*lacZ*4787, *HsdR*514, DE (*araBAD*)567, DE (*rhaBAD*)568, *rph−1*], was used as the WT, which was the parent strain of the KEIO mutant collection [31,32]. *ygaV* and *ygaP* mutants were also obtained from the KEIO collection (JW2642-KC and JW2643-KC, respectively). Before experiments, inserted antibiotic-resistance genes in *ygaV* and *ygaP* were removed as described previously [32]. *E. coli* strains, DH5α and BL21(DE3), were used in plasmid construction and protein expression, respectively. These *E. coli* strains were subsequently grown at 37 °C in LB broth. When necessary, Amp, Sm, and Km were added to the medium at a concentration of 50 mg/L each, except for the pPAP404-containing *ygaV* mutant (described below) for which Amp was added to the agar-solidified medium and liquid medium at a concentration of 50 and 20 mg/L, respectively.

The complementing strain of *ygaV* mutant was constructed as follows. The cording region of *ygaV* was amplified by PCR with a primer pair, 5′-AATTCGAGCTCGGTACCCGGGGATCCATGGATTATAAAGATGATGACGATAAAACTGAACTCGCGCAATTACAG-3′ and 5′-GGCTGCAGGTCGACTCTAGAGGATCCTTACGGACAATAGACATTTTTC-3′.

The genomic DNA of the BW25113 strain was used as a PCR template. The amplified fragment was cloned into the ampicillin-resistant pPAB404 plasmid [33] at its *Bam*HI site. The resulting plasmid was transferred into the above *ygaV* mutant by a standard transformation protocol. The pPAB404 vector was transferred into WT and *ygaV* mutant for growth curve measurement.

### 2.2. RNA Sequencing

*E. coli* WT and *ygaV* mutant were grown overnight in LB medium at 37 °C. The next day, each culture was diluted 20 times using three milliliters of fresh LB medium and grown in test tubes with intermittent shaking. At optical density values between 0.25 and 0.3, one milliliter of each culture was then harvested by centrifugation at 5000× *g* and a temperature of 4 °C. Total RNAs were subsequently extracted using the RNeasy Mini kit (Qiagen, Hilden, Germany), followed by TURBO DNase (Ambino) treatment to eliminate genomic DNA according to manufacturer’s instructions, after which an equal quantity of total RNAs from three biologically independent samples of each strain was mixed for use in RNA sequencing. RIKEN Genesis, Japan conducted the subsequent RNA-sequencing procedures. The libraries were constructed by the TruSeq Stranded total RNA kit (Illumina, San Diego, CA, USA) with NEBNext Bacterial rRNA depletion Kit (NEB) and 150 bp paired-end sequencing was performed on a NovaSeq 6000 (Illumina). The sequencing data were deposited in DDBJ Sequence Read Archive under the accession number DRA012765. RNA-seq reads were mapped to the reference genome using HISAT2 [34]. Mapped reads were counted using htseq-count [35]. Differential gene expression analysis was performed using the TMM normalization method and edgeR test method in the TCC package [36]. The reference genome sequence, GFF file and annotation data of *E. coli* strain BW25113 [37] were obtained from ASAP [38].

### 2.3. Quantitative Real-Time PCR

Total RNA was isolated from *E. coli* WT and *ygaV* mutant grown mid-log phase using the RNeasy Mini kit (Qiagen). Quantitative real-time PCR was performed using the total RNA as templates with PrimeScript RT reagent kit (TaKARa, Kyoto, Japan) on a Thermal Cycler Dice real-time System III (TaKaRa). The level of *gyrA* mRNA was used as an internal standard. The primer pairs used are *narH* (5′-GCGTGGAATATGCGTGGTTC-3′ and 5′-CTTGTAAATCGCACCGCTCG-3′), *dmsA* (5′-GCGAACTGGCAAAACGTCTT-3′ and 5′-TATCGCCGTTGTGGATACCG-3′), *napA* (5′-ACGAAGTGCTGTATGCCACA-3′ and 5′-CTTTCACTTTGTCGCCACGG-3′), *ygaV* (5′-ACTGAACTCGCGCAATTACA-3′ and 5′-GAGGCACTCAGTCCGGTAATG-3′) and *gyrA* (5′-ACCTTGCGAGAGAAATTACAC-3′ and 5′-CATTGCCTAGTACGTTCATGG-3′). The real-time PCR experiment was repeated using three independent biological replicates.

### 2.4. Expression and Purification of YgaV

Polymerase chain reaction (PCR) with the primer pair, YgaV-pCold-F (5′-TCATATCGAAGGTAGGCATATGACTGAACTCGCGCAATT-3′) and YgaV-pCold-R (5′-CAGGTCGACAAGCTTGAATTCTTACGGACAATAGACATTTT-3′), was used to amplify the DNA fragment encoding *ygaV*, using WT genomic DNA as a template. The amplified fragment was then cloned into a pColdI vector (TaKaRa) at *Nde*I and *Eco*RI sites, following the SLiCE method [39]. After checking the correct sequences of the cloned DNA, the resulting plasmid was introduced into the strain BL21(DE3). Afterward, the expression strain for N-terminally His-tagged YgaV was grown in LB medium with Amp at 37 °C. At OD_600_ = 0.6, 0.5 mM, isopropyl β-D-thiogalactopyranoside (IPTG) was added to induce His-tagged YgaV expression, then cells were further incubated at 15 °C for ~16 h. Subsequently, harvested cells were suspended in a buffer, containing 500 mM NaCl, 5 mM imidazole, and 20-mM Tris/HCl (pH 7.9), after which sonification was used to disrupt the cells, followed by centrifugation at 17,000× *g* and a temperature of 4 °C. The supernatant obtained was finally collected and applied to Ni-column chromatography with His-bind resins (Merck, Darmstadt, Germany) according to the manufacturer’s instructions, then purified His-tagged YgaV was analyzed through SDS-PAGE and stained with Coomassie brilliant blue.

### 2.5. Reconstitution of Heme-Bound YgaV

The hemin solution was freshly prepared by dissolving hemin (Sigma-Aldrich, St. Louis, MO, USA) in 0.01 M NaOH. The final hemin concentration obtained was determined spectrophotometrically using a coefficient of 58.4 mM^−1^ cm^−1^. Subsequently, purified His-tagged YgaV was mixed with two-fold molar excess hemin at room temperature for 20 min. Then, free hemin was removed by passing the mixture through a PDMidiTrapTMG-25 column (GE Health care, Chicago, USA) with a buffer, containing 500 mM NaCl and 20 mM Tris/HCl (pH 7.9). Absorption spectra of apo-YgaV and heme-YgaV were finally measured using a Shimazu UV-1800 spectrophotometer (Shimazu, Kyoto, Japan).

### 2.6. Electrophoretic Mobility Shift Assay

PCR was used to amplify the *ygaV* and *katG* operator/promoter regions using two primer pairs; ygaV-pro forward (5′-GAGCTCGGTACCCGGGGATCGATTCTCCAGAAACCCATAT-3′) and ygaV-pro reverse (5′-CAGGTCGACTCTAGAGGATCCCTGTTCGGCACTGGCCTGT-3′) for the *ygaV* promoter, then katG-pro forward (5′-GAGCTCGGTACCCGGGGATCTAAATAGTGTGGCTTTTGTGAAAAT-3′) and katG-pro reverse (5′-CAGGTCGACTCTAGAGGATCCATCAATGTGCTCCCCTCTACAG-3′) for the *katG* promoter. Afterward, each amplified fragment was separately cloned into a linearized pUC19 vector using an In-Fusion HD Cloning Kit (TaKaRa). Next, PCR was conducted again to amplify the inserted DNA of the resulting plasmids using the Cy5.5-labeled forward primer [40] and a non-labeled reverse primer (5′-CAGGAAACAGCTATGA-3′). Finally, each amplified DNA was purified from the agarose gel and used as a probe for the gel mobility shift analysis.

The DNA probe (50 nM) was incubated for 15 min at room temperature in a 10 µL binding buffer composed of 25 mM Tris-HCl (pH 8.0), 100 mM NaCl, 2 mM MgCl_2_, 6% glycerol, and 50 mg/mL heparin. This mixture was then incubated with various concentrations of purified apo-YgaV and Heme-YgaV for 30 min at room temperature. Subsequently, proteins were pretreated with 1 mM DTT or 10 mM Na_2_S to be reduced and oxidized, respectively. Then, 5% PAGE was used to separate samples at room temperature in a buffer composed of 25 mM Tris-HCl (pH 8.0), 1 mM EDTA, and 144 mM glycine. After electrophoresis, gels were analyzed using an image analyzer; Odyssey (LI-COR, Lincoln, MI, USA).

### 2.7. DNase Footprint Assay

PCR provided Cy5.5-labeled DNA fragments of the *ygaV* and *katG* operators/promoters using the labeled primer pair as described above (electrophoretic mobility shift assay). For the analysis of these operators/promoters, each labeled DNA fragment (100 nM) was mixed with purified apo-YgaV at various concentrations in a 22 μL buffer, containing 12.5 mM HEPES/NaOH (pH 8.0), 5 mM K-acetate, 2.5 mM Mg-acetate, 1 mM CaCl_2,_ 12.5 µg mL^−1^ bovine serum albumin, 1 mM DTT, and 0.3 mg mL^−1^ heparin. Next, the mixtures were incubated at room temperature (~22 °C) for 30 min, after which 3 μL of 80 times diluted DNase I (TaKaRa) was added to the reaction mixtures and further incubated for 15 min. The reaction was then stopped by adding 25 µL of 0.5 M EDTA (pH 8.0). DNA fragments obtained were purified using a MinElute PCR purification kit (QIAGEN). Afterward, 0.5 µL of 600 LIZ Size Standard (Applied biosystem) was added to each purified sample (~15 µL), followed by an analysis of samples with a 3730xl DNA analyzer (Thermo Fisher Scientific, Waltham, MA, USA). Finally, fragment analysis was conducted using a Peak scanner software version 1.0 (Thermo Fisher Scientific).

### 2.8. In Vitro AMS Modification

Purified apo-YgaV and heme-YgaV (0.1 mM) were treated with DTT, CuCl_2_, GSSG, Na_2_S, and GSSG plus Na_2_S at different concentrations for variable time intervals, as indicated in each figure legend. Samples were then mixed with 10% (*w*/*v*) trichloroacetic acid (TCA), followed by incubation on ice for 30 min. Later, samples were centrifuged at 15,000× *g*, and the precipitates formed were washed using ice-cold acetone. After repeating the washing step thrice, protein precipitates were resuspended in a buffer containing 0.1% SDS, 0.1 M Tris-HCl (pH 8.0), and 15 mM 4-acetamido-4′-maleimidylstilbene-2,2′-disulfonicacid (AMS) (Molecular Probes, Eugene, OR, USA), then incubated at 37 °C for 2 h. Finally, proteins in the samples were separated using 18% SDS-PAGE, followed by staining with Coomassie brilliant blue.

### 2.9. Ionization-Associated Cleavage of Oxidized Polysulfur (iCOPS) Method

Oxidized polysulfur structure in YgaV was analyzed by iCOPS method. Specifically, recombinant YgaV protein was treated with 10 mM Na_2_S at room temperature for 30 min, then YgaV protein (5 μM) was incubated in 0.25 M phosphate buffer (pH 2.5) containing 2 M urea and 6 μg mL^−1^ pepsin at 37 °C for 4 h in the dark. The mixture was applied onto a reversed-phase cartridge (Discovery^®^ DSC-18, Cat#2602-U) (Supelco, St. Louis, MO, USA), which was pre-equilibrated by 0.1% formic acid (FA). After washing with 0.1% FA, peptides were eluted by 0.1% FA containing 80% methanol, eluates were concentrated *in vacuo*, then were subjected to HPLC-ESI-MS/MS. HPLC-ESI-MS/MS analyses were carried out using a Xevo TQD Triple Quadrupole Mass Spectrometer (Waters, Milford, MA, USA) coupled to an Alliance e2695 HPLC system (Waters). Peptides were separated by the Alliance e2695 system on a Mightysil RP-18 GP column (50 × 2.0 mm inner diameter, Cat#25543-96) (Kanto Kagaku, Tokyo, Japan,) and then eluted using methanol as the mobile phase with a linear gradient (1–99%, 10 min) in the presence of 0.1% FA at a flow rate of 0.3 mL/min at 40 °C. The mass spectrometer was operated in the positive mode with the capillary voltage and desolvation gas (nitrogen) set to 1000 V and 1000 L/h, respectively, at a temperature of 500 °C. Analytes were detected in the multiple reaction monitoring (MRM) mode. The MRM parameters are listed in Appendix A.

For the evaluation of iCOPS method, 10 μM GS-S-SG, as a model compound for oxidized polysulfides, was also analyzed by the same approach.

### 2.10. Primer Extension Analysis

The *E. coli ygaV* mutant was grown in LB medium until the mid-log phase (OD_600_ = 0.3) was harvested, and total RNA was extracted using the RNeasy Mini kit (Qiagen). Reverse transcription was then conducted using QuantiTect Reverse Transcription Kit (QIAGEN) with 7 mg purified RNA and 100 µM of the 6-FAM–labeled primer (5′-GTTCGGCACTGGCCTGTAATTGCGCGAG-3′). The mixture was subsequently incubated at 42 °C for 15 min, then incubated again at 95 °C for 3 min to stop the reaction. Afterward, synthesized cDNA was purified by ethanol precipitation, and resuspended in 12 µL Hi-Di formamide (Thermo Fisher Scientific). Then, 0.5 µL of 600 LIZ Size Standard (Applied biosystem) was added to the purified sample (~15 µL), which was analyzed using a 3730xl DNA analyzer (Applied biosystems). Finally, fragment analysis was conducted with Peak scanner software v.1.0 (Thermo Fisher Scientific).

### 2.11. Construction of lacZ Fusion and β-Galactosidase Assay

The DNA fragment of the *ygaV* operator region (411-bp upstream from the start codon of YgaV) was amplified using PCR, with the primer pair; (5′-CGGAATATTAATAGGTCTAAATTAAGTAAACTCTAAAC-3′) and (5′-CATATGCATCCTAGGTCTTTCACAAAATTGAACAGACCC-3′). The amplified fragment was subsequently cloned into *Stu*I-cut pCF1010 [41], using the In-Fusion HD Cloning Kit (Clonetech). Later, the resulting *lacZ*-fusion plasmid was introduced into *E. coli* WT, *ygaV,* and *ygaP* mutants. Then, the strains were grown in LB medium, after which 0.2 mM Na_2_S (final concentration) was added at the mid-log phase (OD_600_ = ~0.3) and incubated further until the late-log phase (OD_600_ = 0.8~0.9). After induction, one-milliliter culture cells were harvested and suspended in a buffer, containing 50 mM NaH_2_PO_4_, 60 mM Na_2_HPO_4_, 10 mM KCl, 1 mM MgSO_4_, followed by measurement of LacZ activity as described previously [42].

### 2.12. Measurement of the H_2_O_2_ Content

*E. coli* cells grown for ~18 h at 37 °C in LB broth were diluted by 20-fold with fresh LB medium, grown until OD_600_ = 0.5−0.6, and then treated or untreated with 5.0 μg mL^−1^ Km and/or Sm (final concentration) for 5, 10 and 30 min. H_2_O_2_ content was measured as previously described [43]. Briefly, 900 μL aliquots of the cultures were mixed with 100 μL of 100% (*w*/*v*) trichloroacetic acid followed by centrifugation (18,800× *g*, 10 min). A 250 μL aliquot of the supernatant was mixed with the same volume of 10 mM potassium phosphate buffer (pH 7.0) and 1 M potassium iodide. Samples were incubated for 20 min at room temperature in darkness, and then the 390 nm absorption of each sample was measured using the UV-1800 spectrophotometer (Shimadzu, Tokyo, Japan).

### 2.13. Antibiotic Susceptibility Test

*E. coli* cells grown for ~18 h at 37 °C in LB broth were diluted 20-fold with fresh LB medium, grown until OD_600_ = 0.5, and then lawned on LB plates. A paper disk saturated with Amp, Km, Sm and/or Tc (BD BBL sensi-disc; ampicillin 10, kanamycin 30, streptomycin 10, and tetracycline 30, respectively) was then placed on WT, *ygaV* (or *ygaP*) mutant and *ygaV* complementing strain lawns grown under aerobic, anaerobic, or aerobic H_2_S-atmosphere conditions without any antibiotics. For anaerobic growth, plates were put in a sealed plastic bag, containing an oxygen absorber AneroPack-Anaero (Mitsubishi Gas Chemical, Tokyo, Japan). O_2_ levels in the plastic bag were measured by a gas detector OXS-2200 (New Cosmos Electric, Osaka, Japan), which indicated that they were 21.0% before the experiment and less than 0.0% after >1 h incubation. For gaseous-H_2_S atmosphere conditions, plates were put in a sealed plastic jar (*ϕ* = 15 cm, *h* = 20 cm) comprising a small tube with one milliliter 1N HCl-dissolved 10% (*w*/*v*) thioacetamide to generate H_2_S gas as reported previously [24]. O_2_ and H_2_S levels in the jar were measured by a gas detector OXS-2200 (New Cosmos Electric), which indicated that they were 21.0% and 0.0 ppm, respectively, before the experiments, and 19.7% and 10.3 ppm, respectively, after >1 h incubation. We also estimated sulfide levels in the LB medium incubated in the H_2_S-atmospheric conditions. Specifically, 20 mL LB medium was placed in a petri dish and incubated in the H_2_S-generating jar for 5 h. Then, sulfide levels in the LB medium were measured by OxiSelect Free Hydrogen Sulfide Gas Assay Kit (Cell Biolabs, San Diego, CA, USA). Specifically, sulfide concentration in the LB medium was estimated based on a calibration curve prepared with 10, 20, 30, 40, 50 and 60 μM Na_2_S standard solution (prepared by dissolving Na_2_S, included in the kit, with deionized water) according to the manufacturer’s instruction (Appendix A).

Susceptibility to Km and Sm was also tested with liquid cultures as follows. *E. coli* cells grown until OD_600_ = 0.30~0.35 as above were treated with 0.2 μg mL^−1^ or 1.0 μg mL^−1^ Km or Sm (final concentration) for 30 min with or without 0.2 mM Na_2_S (final concentration). The treated cultures were serially diluted to 10^−1^~10^−6^-fold, and each dilution series was spread on an LB plate. After incubation for 18-h at 37 °C under aerobic conditions, susceptibility to Km or Sm was expressed as colony-forming-unit (CFU) of each 1 mL culture.

### 2.14. Statistics and Reproducibility

Statistical significance of data was tested by two-sided Student’s *t*-test and/or Tukey’s test by Excel and MEPHAS http://www.gen-info.osaka-u.ac.jp/MEPHAS/tukey-e.html (accessed on 1 November 2021), respectively. The sample size (*n*) and the nature of replicates have been given wherever relevant.

### 2.15. Accession Numbers

*E. coli* YgaV, P77295; YgaP, P55734. Other accession numbers of *E. coli* genes are available on the EcoCyc website https://biocyc.org/ECOLI (accessed on 1 August 2021).

## 3. Results

### 3.1. Identification of Genes Controlled by YgaV

A previous study identified YgaV-target genes (*ygaV* and *ygaP*) [44]. The *ygaP* gene encodes a rhodanese-like enzyme, which is located downstream of *ygaV* and is cotranscribed with an operon containing *ygaV* in *E. coli*. Whereas the *ygaVP* operon was previously derepressed by tributyltin, and YgaV was required for the repression of this transcription [44]. Therefore, YgaV functioned as a repressor for the operon. To further identify YgaV-regulated genes, we conducted an RNA-sequencing (RNA-seq) transcriptomic analysis of *E. coli* wild-type (WT) and *ygaV* null mutant grown aerobically. The gene’s log-ratio and mean-average of read counts (MA plot) were then constructed based on the RNA-seq transcriptome data obtained, which indicated that 98 and 44 genes were significantly upregulated and downregulated (*p*-value < 0.01), respectively, in the *ygaV* mutant compared to those in the WT (Appendix A). Read counts for the *ygaP* gene region were markedly increased as well in the *ygaV* mutant (Appendix A), which was consistent with a previous study [44]. Hence, we established that YgaV repressed the *ygaVP* operon’s transcription. Interestingly, 35 genes out of the 98 genes upregulated in the *ygaV* mutant were annotated in databases as genes involved in “anaerobic respiration” (Appendix A) that comprised genes for dimethyl sulfoxide reductase (*dms*), nitrate reductases (*nar* and *nap*), formate dehydrogenase (*fdn*), nitrite reductases (*nir* and *nrf*), and fumarate reductase (*frd*). We quantified mRNA levels of *narH*, *dmsA* and *napA* genes by real-time PCR analysis, which showed that transcripts of these genes are significantly increased in the *ygaV* mutant than those in WT (Figure 1), confirming the RNA seq data. In contrast, genes downregulated in the *ygaV* mutant were those encoded by succinate dehydrogenase (*sdh*), proline dehydrogenase (*put*), amino-acid dehydrogenase (*dad*), aldehyde dehydrogenase (*ald*), lactate dehydrogenase (*lld*), and acetolactate synthase (*ilv*) (Appendix A). Notably, read counts for cytochrome *bd* oxidase genes (*cydAB*), whose expression was upregulated through exogenous sulfide [17,19], were also increased in the *ygaV* mutant. Likewise, a catalase gene; *katG*, was upregulated as well in the *ygaV* mutant.

Studies have shown that the redox-dependent regulation of gene expression in *E. coli* was controlled by the well-known global transcription regulator FNR and a two-component system ArcAB [3,4]. The FNR and ArcAB regulons were therefore studied extensively by DNA-microarray analysis, after which we compared the YgaV-regulon (Appendix A) to those of FNR and ArcAB reported previously. Interestingly, the YgaV-regulated genes rarely overlapped with those of FNR or ArcAB. Hence, only eight of the 142 YgaV-regulated genes were controlled through FNR or ArcAB, including *cydA*, *dadX*, *preA*, *nirB*, *frdA*, *sdhA*, *sdhB,* and *gltA* [4,45,46]. These results indicated that YgaV functioned as a distinct transcriptional regulator controlling many anaerobic and ROS-scavenging genes in a different way from that of FNR and ArcAB in *E. coli.*

### 3.2. Biochemical Property of YgaV

The YgaV protein consists of 99 amino acids in length, showing 34% and 24% identities to those of *Rc*SqrR and *Xf*BigR, respectively (Appendix A). YgaV has two Cys residues at the 31st and 98th amino-acid positions (Cys^31^ and Cys^98^), which are conserved among other SqrR/BigR homologs, and those of *Rc*SqrR and *Xf*BigR are required for sensing sulfide [24,27]. For the biochemical analysis of YgaV, His-tagged YgaV was overexpressed in *E. coli*, and purified by Ni-affinity chromatography. SDS-PAGE confirmed that His-tagged YgaV showed high purity (Figure 2A). The isolated YgaV (apo-YgaV) also showed slight absorption at ~500 nm, which was noticeable when the protein was concentrated five times (Figure 2B; red and yellow lines), suggesting the heme binding of YgaV in a way similar to *Rc*SqrR [47]. To confirm whether YgaV binds heme, purified apo-YgaV was mixed with a 2-fold molar excess of hemin [Fe(III) protoporphyrin IX complex], after which reconstituted YgaV (heme-YgaV) was isolated by gel-filtration chromatography. Hemin showed broad doublet peaks at 355 nm and 388 nm (Figure 2B, black lines). However, heme-YgaV showed only at the 355-nm Solet peak, and a new peak appeared at 425 nm (Figure 2B, blue line). Moreover, another broad peak at ~550 nm was clearly observed in heme-YgaV, indicating a 5-coordinated high spin complex. These data confirmed that YgaV is associated with heme like *Rc*SqrR [47].

Next, we checked the DNA-binding property of YgaV to the *ygaVP* promoter. For this analysis, we first determined the transcription start site of the *ygaVP* operon. The primer extension analysis revealed its transcription start site at the 28-bp upstream of the initial YgaV codon (Figure 3A) where possible RNA polymerase σ70-recognition site, −35 and −10, was identified. DNase footprint analysis with purified apo-YgaV showed that it bound near the transcriptional start and RNA polymerase σ-subunit recognition sites, which were consistent with the fact that YgaV repressed transcription of the *ygaVP* operon (Figure 3B). A similar sequence for the deduced *Rc*SqrR-recognition sites (ATTC-N8-GAAT) [24] was also found in the binding area (green boxes), suggesting that YgaV and *Rc*RqrR recognized similar DNA sequences.

### 3.3. Thiol-Modification of YgaV and Its Impact on DNA Binding

Next, we analyzed the redox status of Cys residues by modification of Cys residues using a thiol-modifying agent, 4-acetamido-4′-maleimidylstilbene-2,2′-disulfonic acid (AMS). AMS derivatives only free Cys sulfhydryl groups, after which the sample is subjected to SDS-PAGE. The oxidation of Cys residues would be resistant to AMS modification. So, any AMS modification that retards electrophoretic mobility would reflect the status of free Cys sulfhydryls in YgaV. Before AMS modification, apo-YgaV was treated with dithiothreitol (DTT), CuCl_2_, oxidized glutathione (GSSG), Na_2_S, and GSSG plus Na_2_S. DTT and CuCl_2_ were used as controls to fully reduce and oxidize the Cys residues. As shown (Figure 4A), apo-YgaV treated with and without DTT similarly showed shifted bands, indicating that Cys residues in purified apo-YgaV contained reduced Cys, which can be modified by AMS. In contrast, CuCl_2_ treatment oxidized the two Cys residues as expected, thereby forming an intramolecular disulfide bond. Thus, the Cys residues were unmodified through AMS. Alternatively, GSSG treatment did not show any effects on Cys status, indicating that it did not react with Cys residues. However, Na_2_S treatment, even without GSSG, resulted in resistance to AMS modification, indicating that the sulfide oxidized the two Cys residues in apo-YgaV. Nevertheless, the Na_2_S-dependent modification of Cys residues was dose-dependent. Slight, moderate, and high oxidation of Cys residues were also observed when apo- and heme-YgaV were treated with 0.1, 1, and 10 mM Na_2_S (Figure 4B). Therefore, these results confirmed that a certain concentration of Na_2_S directly oxidized Cys residues of YgaV in vitro.

Electrophoresis mobility shift analysis was then employed to analyze the effects of the Cys modification on the DNA-binding activity of YgaV. The apo-YgaV and heme-YgaV were fully reduced and oxidized by 1 mM DTT and 10 mM Na_2_S, respectively. Afterward, the *YgaVP* promoter DNA probe was mixed. We observed that apo-YgaV and heme-YgaV are dose-dependently bound to the DNA probe (Figure 5A). Binding isotherms indicated that calculated EC_50_ (effective concentration for 50% response) for the binding of Na_2_S-oxidized apo-YgaV and heme-YgaV to the promoter DNA showed approximately five-times increment compared to those of the DTT-reduced ones (Figure 5B), indicating that Cys oxidation by Na_2_S negated DNA-binding of YgaV. However, heme binding to YgaV negligibly affected the DNA binding; EC_50_ of DTT-reduced, but not Na_2_S-oxidized. Furthermore, YgaV showed a ~1.5-fold increment upon heme binding. Similarly, slight effects on DNA affinity by heme binding were also observed for *Rc*SqrR, such that heme slightly inhibited DNA binding of reduced *Rc*SqrR, but not of oxidized (tetrasulfide bond-forming) *Rc*SqrR [47].

The same gel-shift and footprint analysis were conducted with the promoter DNA of *katG*, whose expression was increased in the *ygaV* mutant (Appendix A). As shown (Figure 6A), YgaV bound the *katG* promoter DNA, although a~20-times higher concentration of YgaV was needed for the band-shift compared to those observed for the *ygaV* promoter (Figure 5A). DNase footprint analysis results further showed the YgaV-binding site (Figure 6B). As observed, this binding site was −113 and −157 bp from the transcriptional start site of *katG* (Figure 6C). It has been established that *katG* transcription was activated by the H_2_O_2_-responsive transcriptional regulator OxyR [48]. The OxyR-binding site has been identified to range from −32 to −76 bp upstream of the transcriptional start site (Figure 6C, blue) [48], which overlapped with the −35 motif of the RNA polymerase σ-subunit-recognition site. However, the YgaV-binding site (Figure 6C, red) was located upstream from the RNA-polymerase recognition site, which was in contrast to those observed for the *ygaV* gene, where YgaV binds to the RNA-polymerase recognition site (Figure 3B). Similar sequences of the deduced *Rs*SqrR-recognition sites (ATTC-N8-GAAT) were unidentified in the YgaV-binding area in the *katG* promoter. These results, therefore, indicated that although YgaV is bound to the *katG* promoter, we propose that the mechanism of how YgaV regulates *katG’s* transcription is different from that for the *ygaV* promoter.

### 3.4. Detection of Oxidized Polysulfur Structure of YgaV by iCOPS Method

To investigate the Cys structure of Na_2_S-oxidized YgaV, we developed a novel technology termed ionization-associated cleavage of oxidized polysulfur (iCOPS). The scheme of iCOPS is shown in Appendix A. The iCOPS utilizes the high-performance liquid chromatography-electrospray ionization-tandem mass spectrometry (HPLC-ESI-MS/MS) with low- and high-voltage ionization, which allows us to confirm the existence of polysulfur bond(s) in the target protein (more details, see Materials and Methods). Specifically, we found that the polysulfur bond is highly sensitive to ionization-associated cleavage as examined by a model experiment with glutathione trisulfide (GS-S-SG), which is potentially converted to glutathione (GSH) and GSSH by ionization-associated cleavage. After determining the exact conditions of HPLC-ESI-MS/MS to selectively detect GS-S-SG, GSH and GSSH by multiple reaction monitoring (MRM) mode (Appendix A), we found that GS-S-SG was stably detected under low-voltage (<25 V) ionization conditions; on the other hand, the intact GS-S-SG signal was decreased with increasing cone voltage, correlating with formation of GSH and GSSH (Appendix A). These results indicated that the polysulfur bond between two peptides can be cleaved under high-voltage (>~60 V) ionization conditions, while its intact form remains under low-voltage (<~30 V) ionization conditions.

We utilized the iCOPS method to investigate oxidized Cys modification in YgaV that was prior treated with 10 mM Na_2_S for 30 min. Given that polysulfides are relatively stable under acidic conditions [49,50,51,52], we digested Na_2_S-oxidized YgaV by pepsin at pH 2.5. Then, peptide fragments containing Cys^31^ or Cys^98^ residue were confirmed with HPLC-ESI-MS/MS analyses with MRM mode (Appendix A). Based on relatively high signal intensity, peptide fragments composed of ILC^31^ML and KNVYC^98^P were chosen for further analyses. As shown in Figure 7A, a peptide complex formed by the tetrasulfide cross-link between Cys^31^ and Cys^98^ could be detected under low-voltage ionization conditions (30 V), while under high-voltage ionization conditions (60 V), the signal intensity of the complex was decreased, and KNVYC^98^P peptide, cleaved from the complex, was simultaneously appeared. We further investigated the presence of other (longer or shorter) peptides containing oxidized polysulfur species such as disulfide bonding species; however, signal(s) other than the tetrasulfide cross-link was not observed (Figure 7B), indicating that Na_2_S-oxidized YgaV specifically forms an intramolecular tetrasulfide bond between Cys^31^ and Cys^98^.

### 3.5. YgaV Regulates Transcription in a Sulfide-Dependent Manner

We next tested the promoter activity of the *ygaVP* operon in the absence and presence of Na_2_S. The promoter region of the *ygaVP* operon was fused to the *lacZ* gene, and the construct was subsequently introduced into WT as well as *ygaV* null and complementing strains. Note that the endogenous *lacZ* gene on the chromosome was deleted in the strains used. Furthermore, the complementing strain has the *ygaV*-encoding multicopy plasmid, so that the transcript levels of *ygaV* in the complementing strain were significantly higher (~30-fold) than those in WT as examined by real-time PCR analysis (Figure 8A). As shown in Figure 8B, The LacZ activity in WT grown in the presence of 0.2 mM Na_2_S was 3 times higher than that in the absence of sulfides. On the other hand, the LacZ activity was constitutively high, and sulfide-dependent increment of LacZ activity was not observed in the YgaV mutant, indicating that transcription of the *ygaVP* operon was repressed by YgaV, as reported previously [44], and the sulfide negated the repressor activity of YgaV through Cys modification. The LacZ activity in the *ygaV* complementing strain was not increased by the addition of sulfide, suggesting that highly expressed YgaV (Figure 8A) could repress LacZ expression even in the presence of sulfide.

### 3.6. Phenotype of the YgaV Mutant

To obtain more insights into the physiological significance of YgaV’s function, we characterized the phenotype of the *ygaV* mutant. Given YgaV regulates anaerobic respiratory genes whose expression involves ROS generation, we first quantified H_2_O_2_ levels in the *ygaV* mutant grown under aerobic conditions with or without antibiotics. We normalized H_2_O_2_ levels by optical density (OD), as reported in [53], to compare the results of this and the previous studies. Before antibiotics application, H_2_O_2_ levels in WT, *ygaV* mutant and the complementing strains were not significantly different (Figure 8C). However, H_2_O_2_ levels in the *ygaV* mutant were significantly higher than those in WT and the complementing strain when the cells were exposed to 5.0 μg mL^−1^ kanamycin (Km) for 5, 10 and 30 min, and/or streptomycin (Sm) for 5 min (Figure 8C), indicating the importance of YgaV function for preventing H_2_O_2_ generation under antibiotic stress conditions. Note that reduction in bulk H_2_O_2_ levels with increasing incubation time after Km application (Figure 8C) was reported previously [53].

Next, we tested the growth of these mutants on antibiotic-containing plates since sulfide homeostasis was shown to be required for antibiotic tolerance in *E. coli* [15,17]. Specifically, a paper disk saturated with ampicillin (Amp), Sm, tetracycline (Tc) and Km were placed on Luria Bertani (LB) plates where WT, *ygaV,* or *ygaP* mutants, and the *ygaV* complementing strain were grown. The plates were then incubated under aerobic, anaerobic, and gaseous-H_2_S-atmosphere conditions that mimic the sulfide environment in the gastrointestinal tract where enteric bacteria such as *E. coli* thrive. For anaerobic growth, plates were put into a sealed plastic bag with an oxygen absorber. However, for gaseous-H_2_S conditions, plates were put into a sealed plastic jar with a small tube containing HCl-dissolved thioacetamide, which generates H_2_S gas. Under the H_2_S conditions, H_2_S could be used as an electron donor for *R. capsulatus* to grow photoautotrophically [24]. Notably, we did not place an oxygen absorber into the jar, indicating that the jar’s internal environment was aerobic. In fact, O_2_ and H_2_S levels in the gaseous-H_2_S conditions jar were 19.7% and 10.3 ppm, respectively, after incubation for 18 h, as examined by an electronic gas detector (more details, see Section 2). In addition, under the three growth conditions, *ygaV* mutant and the *ygaV* complementing strain could grow as WT in the absence of antibiotics (Appendix A). As shown in Figure 9, zone borders of WT cells grown under aerobic conditions in the presence of Amp, Sm and Tc were larger than those of WT cells grown under aerobic H_2_S-gaseous conditions, supporting previous studies that sulfide contributes to antibiotic tolerance in *E. coli* [17]. However, the H_2_S-dependent antibiotic tolerance was not obvious for Km. Strikingly, zone borders shown for the *ygaV* mutant were larger than those of WT under both aerobic and aerobic H_2_S-gaseous conditions. These phenotypes were however unobserved in cells grown under anaerobic conditions. Zone borders of the *ygaP* mutant on Amp-containing plates, and the complementing strain on Sm, Km and Tc were also similar to those of the WT under all growth conditions tested. Note that we could not test the growth of the complementing strain on Amp plates because the stain harbors an ampicillin-resistance plasmid.

We also checked the effects of sulfide on the survival of *E. coli* cells exposed to Sm and/or Km in liquid culture. We observed ~1.5-fold increment in colony-forming-unit (CFU) of WT exposed to 0.2 μg mL^−1^ Km for 30 min, and 0.2 μg mL^−1^ and 1.0 μg mL^−1^ Sm in the presence of 0.2 mM Na_2_S (Figure 8D). This phenotype was not observed in the *ygaV* mutant, indicating that YgaV-dependent transcriptional regulation contributes to antibiotic tolerance. The sulfide effect was not observed when cells were treated with 1.0 μg mL^−1^ Km (Figure 8D). WT did not show clear increased tolerance to Km under H_2_S-gaseous conditions as examined by the disk assay (Figure 9), suggesting that the influence of sulfide on antibiotic tolerance is dependent on antibiotics. The sulfide effect on antibiotic tolerance was observed in the complementing strain, although the strain showed constitutive higher CFU for Sm.

We further tested weakly antibiotic tolerance of *ygaV* mutant. Specifically, the Amp-resistance empty vector was transferred into the WT and *ygaV* mutant and we checked their tolerance to Amp. Surprisingly, although all recombinant colonies could be selected on agar-solidified plates containing 50 μg mL^−1^ Amp, overnight culture of the *ygaV* mutant harboring the empty vector could not be obtained when a colony was placed into a liquid medium containing the same amount of Amp (50 μg mL^−1^) (Figure 10A). Overnight cultures of the strains grown in liquid medium containing 20 μg mL^−1^ were diluted 20 times with fresh liquid medium containing 80 and/or 20 μg mL^−1^ Amp, and then growth was monitored (Figure 10B). The growth of the *ygaV* mutant, but not WT, harboring the empty vector, was inhibited and lowered in a liquid medium containing 80 μg mL^−1^ and 20 μg mL^−1^ Amp, respectively, although the mutant showed similar growth to those of WT and the complementing strain until OD600 = ~0.3. These results further indicate the importance of YgaV function for antibiotic tolerance.

## 4. Discussion

In this study, we conducted the genetic and biochemical characterization of the SqrR/BigR homolog in *E. coli*; YgaV. RNA-seq analysis revealed that YgaV negatively regulated many anaerobic respiratory genes, including formate, fumarate, lactate, nitrate, and nitrite reductase genes, most of which were not controlled via the well-known global transcription regulators; FNR and a two-component system ArcAB [45,46]. Furthermore, YgaV functioned as both a repressor and an activator such as FNR and ArcAB (Appendix A), suggesting the importance of YgaV for improving expression dynamics of aerobic/anaerobic respiratory genes. In the *ygaV* mutant, many genes involved in ROS scavenging are also significantly upregulated under oxic conditions, suggesting the significance of the regulatory networks of the transcription factors to grow in the gut where oxic/anoxic conditions are drastically interchanged. It should be noted that genes regulated by FNR and ArcAB were previously characterized by microarray analysis, although we used RNA seq analysis to find YgaV-regulated genes. Thus, the relevance of the comparison of the data obtained by different techniques should be taken into account in consideration.

The next question was how the sulfide sensitivity of YgaV influenced *E. coli’s* physiology. Cys modification of YgaV was therefore observed with ~10 mM Na_2_S in vitro (Figure 4). Given Na_2_S is in equilibrium with hydrosulfide ion in solution, and the deduced sulfide concentration was nearly identical to those in the gut (0.2–2.4 mM) [13], suggesting that YgaV may sense extracellular sulfide to control gene expression. However, because the kinetics of the equilibrium has not been studied, the physiological relevance of the sulfide sensing by YgaV still needs further investigation. Furthermore, WT *E. coli* shows increased antibiotic tolerance under 10.3 ppm H_2_S-atmospheric conditions (Figure 9); the H_2_S concentration in the atmosphere could be calculated to be ~0.5 μM, which is significantly lower than those of Na_2_S (0.2 mM) in the liquid medium (Figure 8D). In fact, sulfide levels in the LB medium, incubated in the gaseous-H_2_S conditions jar were estimated to be less than 10 μM (Appendix A). Although the cells’ sensitivity to gaseous H_2_S and sulfide ion in the liquid medium must be different, these results suggest that antibiotic tolerance caused by low concentration of H_2_S in the air and high concentration of sulfide in the liquid are different, such that they could be due to growth stimulation [23] and reduced antibiotics toxicity [17,19,20,21], respectively. Under sulfide-stress conditions, *E. coli* upregulated the expression of cytochrome *bd* oxidase to switch the terminal oxidase from sulfide-sensitive *bo* type [17,19]. Alteration of the terminal oxidases from the *bo* type to the *bd* type thus inhibited ROS production and stimulate growth that contributes to antibiotic tolerance [17,19,20,21]. Nevertheless, the mechanism of the sulfide-dependent upregulation of cytochrome *bd* oxidase expression remained unknown. Our RNA-seq data indicated that YgaV controlled the transcription of cytochrome *bd* oxidase genes and that of a catalase gene *katG* (Figure 6; Appendix A). These results obtained suggested the physiological importance of sulfide-dependent transcription regulation by YgaV, as it adjusted respiratory growth modes, redox balance, ROS production, and antibiotic tolerance (Figure 11). Inhibited growth of the *ygaV* mutant, caused by Amp addition, was specifically observed in the latter growth stage (Figure 10B), where the terminal oxidase might be switched from the *bo* type to the *bd* type to adapt to lowered oxygen levels [54], further suggesting the importance of the YgaV-dependent transcriptional regulation for antibiotic tolerance. It should be noted that H_2_O_2_ levels in the strains after the addition of antibiotics decreased or remained almost constant over time (Figure 8C), suggesting that generated H_2_O_2_ is not directly related to sensitivity to antibiotics. This result is in agreement with previous studies that showed that cell death caused by antibiotics does not depend on ROS [53,55], although involvement of ROS other than H_2_O_2_ still potentially contributes to antibiotic-mediated killing in bacteria [56]. Further research on YgaV may contribute to solving the decade-old enigma of whether there is a relationship between ROS generation and antibiotic tolerance.

However, the above hypothesis contains inconsistencies with the phenotype of *ygaV* mutant. Specifically, highly expressed *katG* and cytochrome *bd* oxidase genes in the *ygaV* mutant may contribute to ROS scavenging [17]; however, H_2_O_2_ levels in the mutant were the same or higher than those in WT before and after antibiotic application (Figure 8C). These results suggest that coordinated regulation of transcriptional repression/activation of not only *katG* and cytochrome *bd* oxidase genes, but also other ~150 YgaV-target genes (Appendix A) is important for ROS scavenging. It is also unclear why sulfide-dependent enhancement of antibiotic tolerance was different for each antibiotic (Figure 8D and Figure 9). Specifically, a sulfide-dependent increase in CFU of WT was observed at 0.2 μg mL^−1^ Km, but not observed at 1.0 μg mL^−1^ Km; on the other hand, it was observed at both 0.2 μg mL^−1^ and 1.0 μg mL^−1^ Sm (Figure 8D). We found that the concentration dependence of antibiotic-induced CFU reduction in WT *E. coli* was greater for Km than for Sm (Appendix A), suggesting that sulfide-dependent enhancement of antibiotic tolerance is observed under sublethal concentrations of antibiotics, the extent of which may vary with each antibiotic. Furthermore, it is also unclear why *ygaV* mutant showed lower antibiotic tolerance than WT on plates under sulfide-free conditions (Figure 9, aerobic), although WT and *ygaV* mutant showed similar tolerance to antibiotics in sulfide-free liquid-medium (Figure 8D). One possibility is that endogenously synthesized H_2_S contributes to antibiotic tolerance in WT, but not in *ygaV* mutant on plates where membrane-permeable H_2_S gas could easily penetrate other cells [57], and the H_2_S-dependent phenotype was not observed in a liquid culture where H_2_S is converted to hydrosulfide ion. In any case, *ygaV* mutant did not show the sulfide-dependent enhancement of antibiotic tolerance (Figure 8D and Figure 9). A previous study identified inhibitors of sulfide synthesis in bacteria, which were recently shown to potentiate antibiotics [58]. In the study, the identification of inhibitors for YgaV was revealed to be important for the discovery of new drug targets.

In our study, YgaV repressed the transcription of *ygaVP* operon, which was derepressed by sulfide (Figure 8B). The negative feedback regulation of the operon should therefore allow strict control of intracellular YgaV levels, which is proposed to be required for the rapid alteration of growth modes upon transition from sulfide to nonsulfide-stress conditions, and vice versa. The expression of YgaP was also synchronized with YgaV’s expression, suggesting the importance of the protein in sulfide-dependent growth controls. However, although YgaP showed rhodanese activity in vitro [59], the exact substrates of YgaP in vivo are unknown, which needs to be clarified in future studies.

Studies have shown that the Cys-dependent reactivity of YgaV was similar to that of *Xf*BigR, causing both to make an intramolecular crosslink upon sulfide treatment [26,27], although *Xf*BigR forms a disulfide bond [26] whereas YgaV forms a tetrasulfide bond in vitro (Figure 7). This suggested that Cys reactivity of YgaV was different from that of *Xf*BigR, although the mechanisms of how a disulfide or tetrasulfide crosslink could be formed by sulfide are still unclear. As shown, *Rc*SqrR did not make any crosslink(s) upon sulfide treatment and required RSS molecules, such as GSSH to form an intramolecular tetrasulfide bond [24]. RSS has also been shown to be generated by cystenyl-tRNA synthase and several sulfur transferases during H_2_S and cysteine metabolic processes [49,60]. Therefore, given that RSS possesses both nucleophilic and electrophilic characteristics, they can control various enzymatic activities involved in various cellular processes, such as redox homeostasis, virulence, and antibiotic resistance in several bacteria, especially in their mitochondria [7,60,61,62,63]. Likewise, our findings revealed that Cys residues of YgaV were also modified in vivo upon sulfide application. Specifically, we observed that 0.2 mM Na_2_S can derepress *ygaVP’s* transcription (Figure 8B), although the Na_2_S concentration cannot completely modify Cys crosslinks of YgaV in vitro (Figure 4), thereby supporting the hypothesis that RSS modifies Cys residues of YgaV in vivo. The fact that ~10 ppm atmospheric H_2_S affects physiology of *E. coli* (Figure 9) and *R. capsulatus* [24] suggests that YgaV and *Rc*SqrR sense exogenous H_2_S in a similar manner.

The results from this study further revealed that *katG’s* gene expression was negatively regulated by YgaV (Appendix A). YgaV’s binding site was detected far upstream from the RNA polymerase-recognition site, and YgaV showed a relatively low affinity to the *katG* promoter in vitro (Figure 6), suggesting that it interacted with another regulator to control *katG’s* expression. One possibility was that YgaV controlled *katG’s* expression by modulating OxyR-binding to the promoter in vivo. Recently, it was found that OxyR can sense RSS and control the expression of enzymes involved in RSS removal, including thioredoxins and glutaredoxins [64]. Given that the intracellular generation of H_2_S and RSS is closely related [60,62], one possibility is that intracellular H_2_S and RSS possessed different signaling molecules in *E. coli*, which was differentially sensed by YgaV and OxyR. Hence, YgaV and OxyR are proposed to have evolved to specifically respond to sulfide and RSS, respectively. Moreover, YgaV did not regulate the OxyR target genes; thioredoxins and glutaredoxins [64] (Appendix A), supporting the hypothesis above.

Our in vitro data revealed that YgaV binds heme as shown in the YgaV homolog *Rc*SqrR (Figure 2B) [47]. One of the conserved Cys residues corresponding to Cys^31^ of YgaV that makes crosslink upon RSS treatment in *Rc*SqrR is involved in the heme-coordination [47], suggesting that YgaV also binds heme in a similar manner. What is the physiological meaning of heme binding? Shimizu et al. proposed that heme-sensing by *Rc*SqrR is important for scavenging RSS rapidly in the sulfide-stress conditions since heme inhibits DNA-binding of *Rc*SqrR to de-repress transcription of some genes involved in RSS metabolism [47]. This may be true in *E. coli*, since a rhodanese YgaP is regulated by YgaV. However, not like *Rc*SqrR, heme did not greatly influence DNA-binding activity of YgaV in vitro (Figure 5B), suggesting different impacts of heme-binding to transcriptional regulation by YgaV and *Rc*SqrR. One possibility is that heme influences the affinity between YgaV and other transcription factors such as OxyR to control gene expression. Clearly, further analysis of YgaV is important for further understanding H_2_S/RSS signaling and its function in bacteria.

## Figures and Tables

**Figure 1 antioxidants-11-02359-f001:**
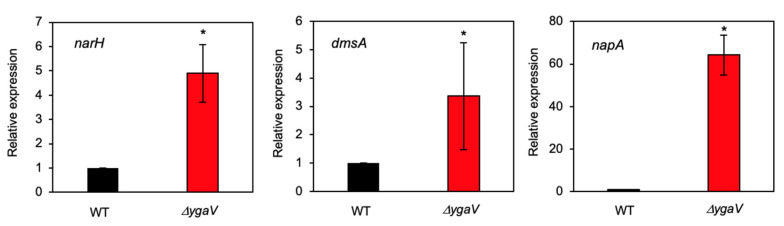
Relative mRNA levels of *narH*, *dmsA* and *napA* genes in WT and *ygaV* mutant (*ΔygaV*) as determined by quantitative real-time PCR analysis. The WT levels were set to 1.0. The *gyrA* gene was used as an internal standard. The values are means ± SD of three biological replicates. ** p* < 0.05, *t*-test.

**Figure 2 antioxidants-11-02359-f002:**
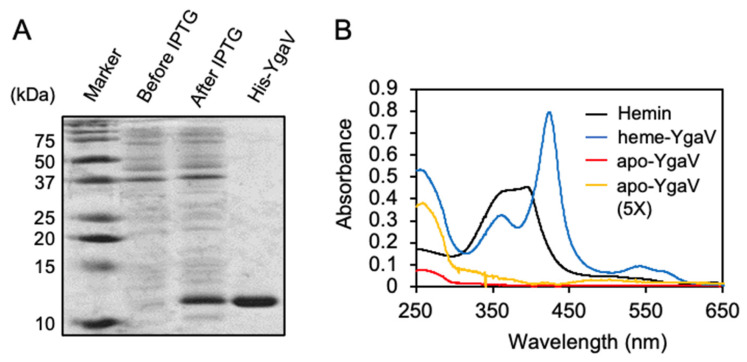
Purification and reconstitution of YgaV: (**A**) SDS-PAGE analysis of purified His-tagged YgaV. Cell extracts of *E. coli* expressing N-terminally His-tagged YgaV, before and after IPTG induction, are also analyzed. Proteins were stained with Coomassie brilliant blue; (**B**) Absorption spectra of hemin, heme-reconstituted YgaV (heme-YgaV), purified YgaV before reconstitution (apo-YgaV), and 5-times concentrated purified YgaV before reconstitution (apo-YgaV, 5×).

**Figure 3 antioxidants-11-02359-f003:**
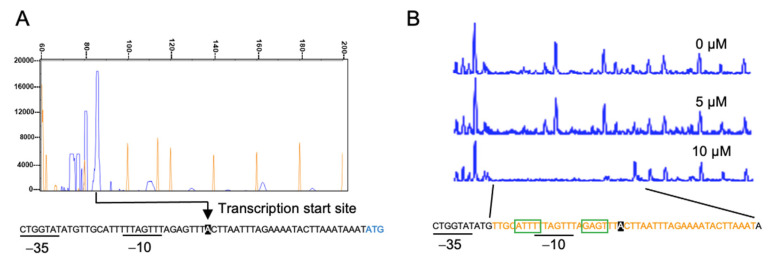
Characterization of the *ygaVP* promoter: (**A**) Primer extension analysis of the *ygaVP* promoter. Signal of the reverse transcripts and size markers are blue and red, respectively. ((**A**), bottom) The RNA polymerase σ-subunit recognition site (−10 and −35) and the translational start site of YgaV (blue) are shown. Reverse transcription was conducted with purified RNA and the fluorescent–labeled primer. The synthesized cDNA was mixed with the 600 LIZ Size Standard (Applied biosystem), and then analyzed using a 3730xl DNA analyzer (Applied biosystems). Finally, fragment analysis was conducted with Peak scanner software v.1.0 (Applied biosystem); (**B**) DNase I footprint analysis of YgaV. Binding to the *ygaVP* promoter DNA with different concentrations of reduced YgaV. Regions corresponding to the DNase I protection regions are shown in red. Motifs similar to the putative *Rc*SqrR-recognition sequences are indicated by green boxes (for details, see text).

**Figure 4 antioxidants-11-02359-f004:**
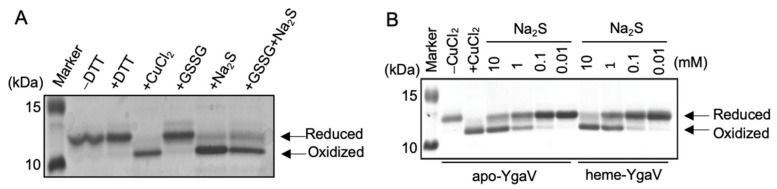
Mobility shifts of YgaV caused by thiol modification on SDS-PAGE gels: (**A**) The purified apo-YgaV was treated with 1 mM DTT, 0.5 mM CuCl_2_, 1 mM GSSG, and/or 10 mM Na_2_S. The proteins were precipitated by trichloroacetic acid (TCA) treatment, labeled with AMS, and then applied for SDS-PAGE; (**B**) The AMS modification was performed with apo- and heme-reconstituted YgaV with different concentrations of Na_2_S. The apo-YgaV was also treated with 0.5 mM CuCl_2_ as a control. Proteins were stained with Coomassie brilliant blue.

**Figure 5 antioxidants-11-02359-f005:**
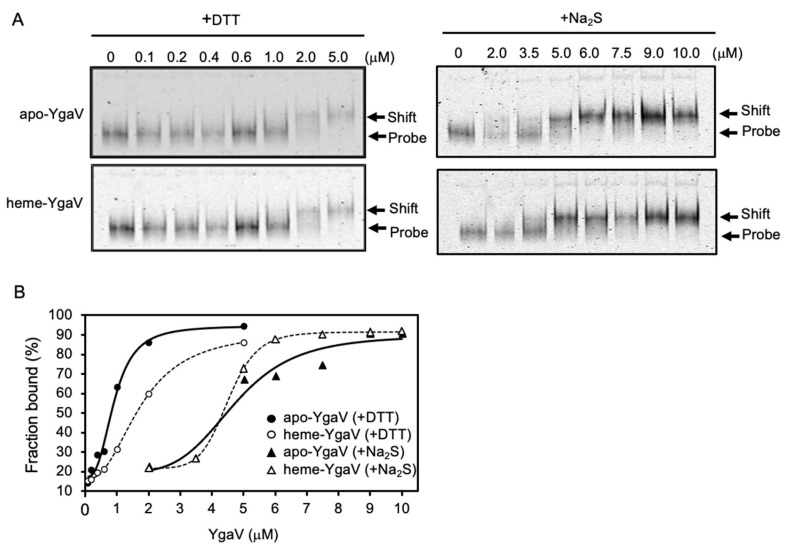
DNA-binding characteristics of YgaV: (**A**) Gel mobility shift analysis of DTT- and Na_2_S-treated apo- and heme-reconstituted YgaV to the *ygaVP* promoter DNA. The fluorescent-labeled DNA probe (50 nM) was incubated with various concentrations of purified apo-YgaV and Heme-YgaV for 30 min at room temperature. Subsequently, proteins were pretreated with 1 mM DTT or 10-mM Na_2_S to be reduced and oxidized, respectively. Then, 5% PAGE was used to separate samples. After electrophoresis, gels were analyzed using an image analyzer; (**B**) Binding isotherms of DTT-treated apo- and heme-reconstituted YgaV, and Na_2_S-treated apo- and heme-reconstituted YgaV.

**Figure 6 antioxidants-11-02359-f006:**
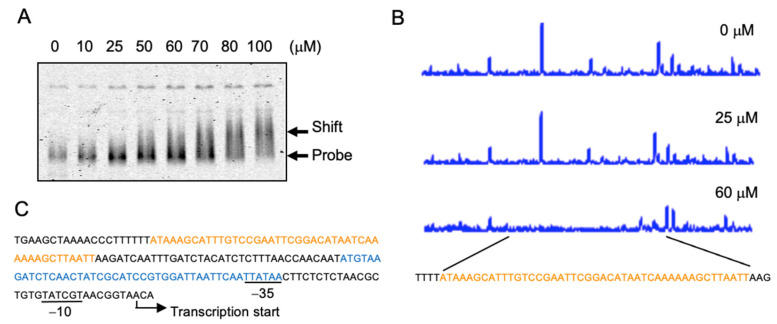
YgaV binding to the *katG* promoter: (**A**) Gel mobility shift analysis of DTT-treated apo-YgaV to the fluorescent-labeled *katG* promoter DNA. The protein-DNA mixtures were separated by 5% PAGE. After electrophoresis, gels were analyzed using an image analyzer; (**B**) DNase I footprint analysis of DTT-reduced apo-YgaV with the *katG* promoter DNA. Regions corresponding to the DNase I protection regions are shown in red. (**C**) The DNA sequence of the *katG* promoter region. OxyR and YgaV-binding sites are indicated by blue and red, respectively. The transcription start site and the RNA polymerase recognition sites (−10 and −35) are also shown.

**Figure 7 antioxidants-11-02359-f007:**
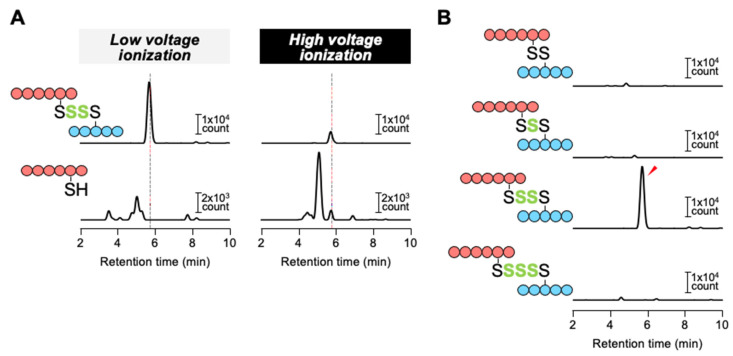
Detection of oxidized polysulfur structure of Na_2_S-treated YgaV by iCOPS method: (**A**) The complex of ILC^31^ML (blue) and KNVYC^98^P (red) with an intra-tetrasulfide bridge was detected under low voltage ionization conditions (30 V), while the oxidized polysulfur bond was disrupted by high voltage ionization (60 V), resulting in the formation of free KNVYC^98^P peptide which was detected at the same retention time as the intact peptide complex (dashed line); (**B**) Various lengths of oxidized polysulfur bridges between ILC^31^ML and KNVYC^98^P peptides were investigated under low-voltage ionization conditions (30 V). A significant signal of the intra-tetrasulfide bridge-containing complex was only detected (red arrow).

**Figure 8 antioxidants-11-02359-f008:**
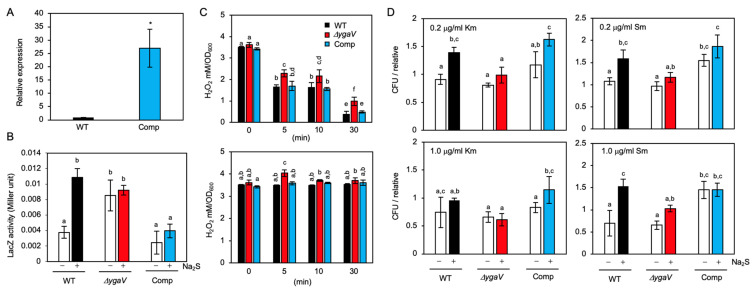
Phenotypes of the complementing strain of the *ygaV* mutant (Comp), which harbors the multicopy plasmid encoding *ygaV* gene: (**A**) Relative mRNA levels of *ygaV* in WT and the *ygaV* complementing strain as determined by quantitative real-time PCR analysis. The WT levels were set to 1.0. The *gyrA* gene was used as an internal standard. The values are means ± SD of three biological replicates. ** p* < 0.05, *t* test; (**B**) Promoter activities of the *ygaVP* operon. LacZ (β-galactosidase) activity measurement with the *ygaV-lacZ* fusion in WT, *ygaV* mutant and the complementing strain. Cells grown at mid-log phase (OD_600_ = ~0.3) were treated or untreated with 0.2 mM Na_2_S (final concentration) and harvested at late-log phase (OD_600_ = 0.8~0.9). The values are means ± SD of three biological replicates; (**C**) H_2_O_2_ levels in WT, *ygaV* mutant and the complementing strain. Cells grown at mid-log phase (OD_600_ = 0.5–0.6) were treated with 5.0 μg mL^−1^ kanamycin (Km) (upper) or streptomycin (Sm) (bottom) (final concentration) for 5, 10 and 30 min, and then harvested. The values are means ± SD of three biological replicates; (**D**) Mean colony-forming-unit (CFU) values of WT, *ygaV* mutant and the complementing strain exposed to 0.2 μg mL^−1^ or 1.0 μg mL^−1^ Km or Sm (final concentration) for 30 min in the presence or absence of 0.2 mM Na_2_S (final concentration). One of the CFU values of WT at 0.2 μg mL^−1^ Km or Sm was set as 1. Different letters (a, b, c, d, e and f) indicate significant differences between groups (*p* < 0.05; Tukey’s test).

**Figure 9 antioxidants-11-02359-f009:**
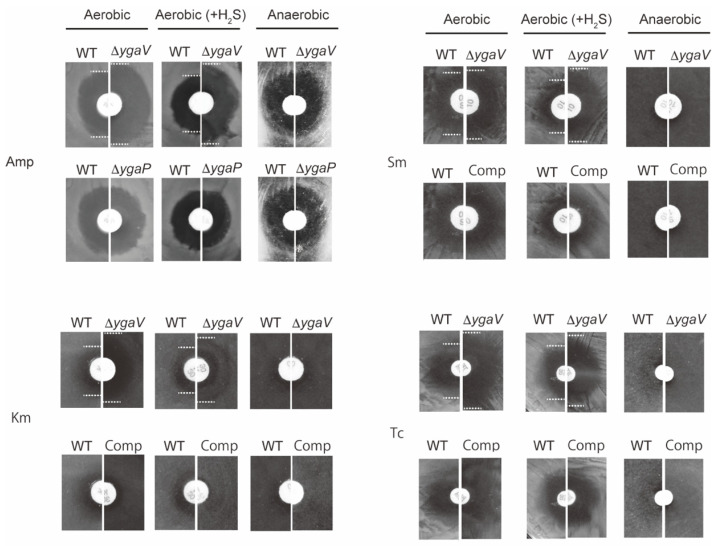
Phenotypic characterization of *ygaV* and *ygaP* mutants. A paper disk saturated with Amp, Km, Sm and Tc was placed on *E. coli* WT, *ygaV* complementing strain (Comp), *ygaV* and/or *ygaP* mutants’ lawns that were grown under aerobic, anaerobic, and aerobic H_2_S-atmosphere conditions for 18 h. Zone borders are marked with dotted lines to easily see the differences.

**Figure 10 antioxidants-11-02359-f010:**
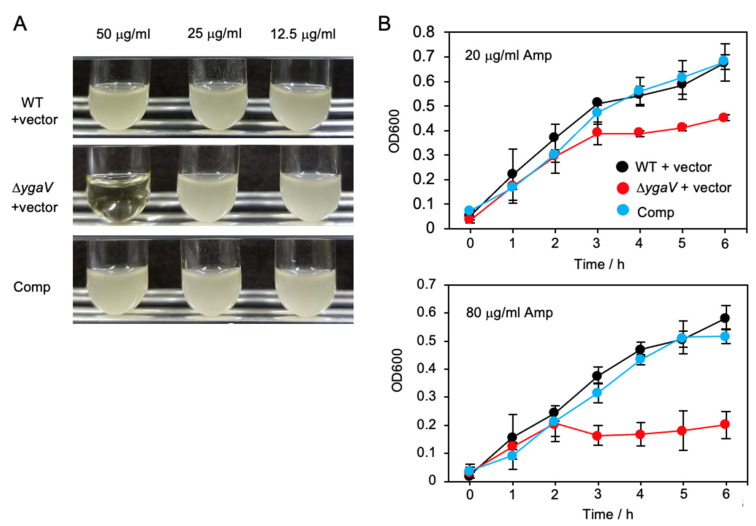
Inhibited growth of the *ygaV* mutant by Amp: (**A**) The Amp-resistance empty pPAB404 vector was introduced into WT and *ygaV* mutants. The complementing strain of *ygaV* mutant (Comp) harbors the pPAB404 encoding *ygaV*. Colonies of each strain on agar-solidified plates containing 50 μg mL^−1^ Amp were picked, placed in liquid medium containing 50, 25 and 12.5 μg mL^−1^ Amp, and then incubated for 16 h; (**B**) Overnight cultures of the three strains shown in (**A**) in liquid medium containing 20 μg mL^−1^ Amp, were diluted 20-times with fresh LB medium containing 80 and 20 μg mL^−1^ Amp, and then growth of the strains was monitored by measuring optical density at 600 nm (OD600).

**Figure 11 antioxidants-11-02359-f011:**
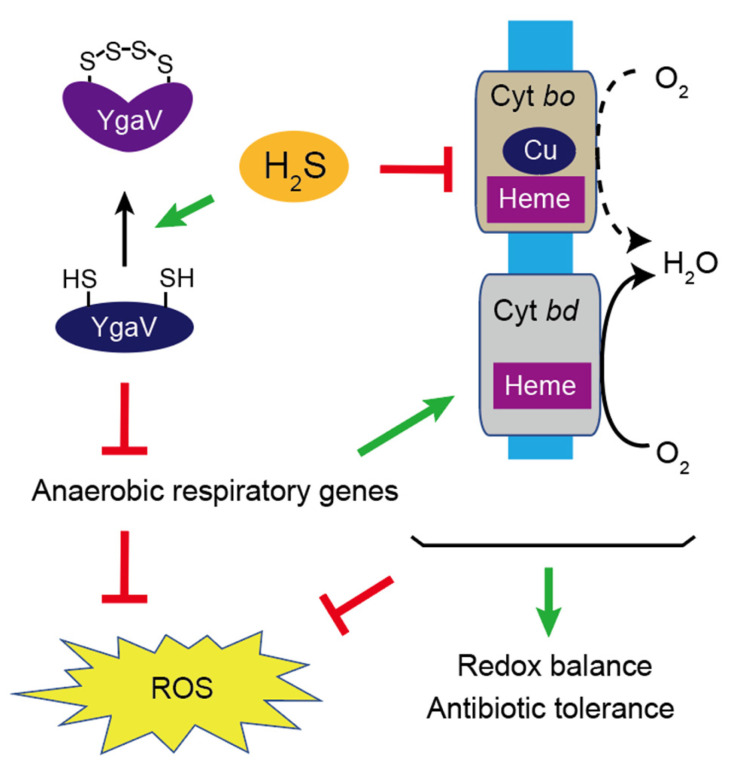
A schematic model for YgaV-dependent control of gene expression and its physiological function. H_2_S inhibits activity of the energy-efficient cytochrome *bo* oxidase, which causes ROS accumulation. In the presence of H_2_S, two cysteine residues of YgaV form an intramolecular tetrasulfide bond to derepress anaerobic-respiratory and ROS-scavenging genes, which contributes to redox homeostasis, ROS scavenging and antibiotic tolerance.

## Data Availability

Data are contained within the article and Appendix A.

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
