# Peer review of "The Sulfide-Responsive SqrR/BigR Homologous Regulator YgaV of Escherichia coli Controls Expression of Anaerobic Respiratory Genes and Antibiotic Tolerance"

_antioxidants, 2022, doi:10.3390/antiox11122359_

Round 1

Reviewer 1 Report

The manuscript of Balasubramanian and coworkers describes a sulfide-responsive transcription factor SqrR/BigR homolog, YgaV of E. coli indicates the regulation of anaerobically respiratory gene expression responding to sulfide compounds, and also scavenging ROS. The ygaV mutant of E. coli did not show the resistance to antichaotic under H2S-atomospheric conditions, but wt and complementary strain of ygaV gene increased the tolerance to them. And also, antibiotic sensitivity of the mutant showed higher than that of wt in the presence and absence of exogenous H2S. All in vivo and in vitro data suggested that the regulation by YgaV was responsible for maintaining redox homeostasis, ROS scavenging, and antibiotic tolerance in E. coli. The manuscript is well written and the results support the conclusions. However, minor adjustment to the manuscript is required for acceptance for publication. Comment for the authors follow:

 1) The use of red and blue colors in Fig. 3 confuses the reader because of the different intensity of the colors in panels A and B. A and B should have different colors or uniform intensity. 

2) In Fig. 10B, it is strange that the early growth is unchanged between wt and the ygaV mutant, ant the latter is changed. Authors should explain this.

Author Response

Reviewer 1

The manuscript of Balasubramanian and coworkers describes a sulfide-responsive transcription factor SqrR/BigR homolog, YgaV of E. coli indicates the regulation of anaerobically respiratory gene expression responding to sulfide compounds, and also scavenging ROS. The ygaV mutant of E. coli did not show the resistance to antichaotic under H2S-atomospheric conditions, but wt and complementary strain of ygaV gene increased the tolerance to them. And also, antibiotic sensitivity of the mutant showed higher than that of wt in the presence and absence of exogenous H2S. All in vivo and in vitro data suggested that the regulation by YgaV was responsible for maintaining redox homeostasis, ROS scavenging, and antibiotic tolerance in E. coli. The manuscript is well written and the results support the conclusions. However, minor adjustment to the manuscript is required for acceptance for publication. Comment for the authors follow:

1) The use of red and blue colors in Fig. 3 confuses the reader because of the different intensity of the colors in panels A and B. A and B should have different colors or uniform intensity. 

OUR ANSWER: We agree. However, the colors of the peak patterns are original ones of each computer application/software, so that we could not change the colors by ourself in the figures. Instead, the red color of the sequence shown in Fig. 3B could be changed, so that we changed the color to be similar to those for Fig. 3A, as suggested.

2) In Fig. 10B, it is strange that the early growth is unchanged between wt and the ygaV mutant, ant the latter is changed. Authors should explain this.

OUR ANSWER: We newly explained the fact that that the mutant showed inhibited growth only in a letter growth stage as suggested (L740~744). Physiological meaning of this result is also discussed in the Discussion section (L802~806).

Reviewer 2 Report

SqrR/BigR proteins work as gene expression regulators and sense H2S (Na2S). The authors characterized the SqrR/BigR homolog protein in E. coli, YgaV, and demonstrated that YgaV senses H2S/Na2S and regulates expression of genes for anaerobic respiration and antibiotic resistance. Characterization of YgaV regulation was performed with traditional and newly developed methods using the purified YgaV protein and the deletion and complementation mutants of ygaV. The work is comprehensive, and the manuscript is carefully written and should be published with minor modifications.   

How/why does Na2S directly oxidize YgaV? Even if it remains unclear, please state to some extent. 

Why is ROS generated in the presence of H2S (e.g., caption for Fig.11)? H2S would directly scavenge oxygen.

Line 553. Given polysulfides are relatively stable -> Given that polysulfides are relatively stable

Lines 576-580. The sentence is ambiguous. Please modify it for clarity or divide it into a few sentences.

Line 654-656.  Cite Figure(s). Line 655. aerobic condition -> oxic condition

Fig. 7B. How do you obtain these standards?

Supplemental Fig. S4. The figure and its caption are not very clear. Because this is to explain the new method the authors developed, the caption for the figure should be more thorough and careful. 

Use of colon and semi-colon seems upside down.

Author Response

Reviewer 2

SqrR/BigR proteins work as gene expression regulators and sense H2S (Na2S). The authors characterized the SqrR/BigR homolog protein in E. coli, YgaV, and demonstrated that YgaV senses H2S/Na2S and regulates expression of genes for anaerobic respiration and antibiotic resistance. Characterization of YgaV regulation was performed with traditional and newly developed methods using the purified YgaV protein and the deletion and complementation mutants of ygaV. The work is comprehensive, and the manuscript is carefully written and should be published with minor modifications.   

1) How/why does Na2S directly oxidize YgaV? Even if it remains unclear, please state to some extent. 

OUR ANSWER: We newly added sentence mentioning that mechanism of how such a crosslink could be formed by sulfide is unckear yet (L850~852).

2) Why is ROS generated in the presence of H2S (e.g., caption for Fig.11)? H2S would directly scavenge oxygen.

OUR ANSWER: The proposed mechanism of how H2S generates ROS is mentioned in the Introduction section (L58~63).

3) Line 553. Given polysulfides are relatively stable -> Given that polysulfides are relatively stable

OUR ANSWER: We corrected it in the revised manuscript (L629).

4) Lines 576-580. The sentence is ambiguous. Please modify it for clarity or divide it into a few sentences.

OUR ANSWER: We agree, and corrected the sentences (L651~670). We thank the reviewer for pointing out this.

5) Line 654-656.  Cite Figure(s). Line 655. aerobic condition -> oxic condition

OUR ANSWER: We added citation of figure (Supplemental Figure S1) (L751~752). We also corrected the wording to “oxic” (L754).

6) Fig. 7B. How do you obtain these standards?

OUR ANSWER: We added sentence explaining how we prepared the standard solution (L441~442).

7) Supplemental Fig. S4. The figure and its caption are not very clear. Because this is to explain the new method the authors developed, the caption for the figure should be more thorough and careful. 

OUR ANSWER: We agree. Because detail of methods and data description were shown in the text, we added a sentence mentioning that “For more details, see Materials and Methods section of the text” for readers.

8) Use of colon and semi-colon seems upside down.

OUR ANSWER: We have corrected the use of colon and semi-colon in the revised text (e.g., L31, L71, L80, L92, L112, L116, L489, L490, L541, L581, L750).

Reviewer 3 Report

The manuscript submitted for publication to Antioxidants by  Balasubramanian et al., titled: "The sulfide-responsive SqrR/BigR homologous regulator YgaV of Escherichia coli controls expression of anaerobic respiratory genes and antibiotic tolerance" is an interesting in vitro work investigating responses to conditions in EColi that control anaerobic respiratory genes and antibiotic tolerance.

The manuscript is well written and structured and easy for the reader to follow. The reviewer would like to bring a couple of points to the attention of the authors so that they take them into consideration.

One important aspect in the discussion section is the potential approach and possible application in the context of the gut microbiome and/or the gut health and responses in the case of exposure exposure. The reviewer would highly suggest the inclusion of a short paragraph in the discussion section discussing potential implication of the findings as well as a broader discussion on gut health and overall health implications in humans. A couple of papers that may be useful to that end for such a discussion are listed below:

  1. Sikalidis AK, Maykish A (2020) The Gut Microbiome and Type 2 Diabetes Mellitus; discussing a complex relationship. Biomedicines. 8(1):8. doi.10.3390/biomedicines8010008.
  2. Sikalidis AK (2015) Amino Acids and Immune Response: A role for cysteine, glutamine, phenylalanine, tryptophan and arginine in T-cell function and cancer? Pathol Oncol Res21(1):9-17. doi: 10.1007/s12253-014-9860-0.

Good work overall.

Author Response

Reviewer 3

The manuscript submitted for publication to Antioxidants by  Balasubramanian et al., titled: "The sulfide-responsive SqrR/BigR homologous regulator YgaV of Escherichia coli controls expression of anaerobic respiratory genes and antibiotic tolerance" is an interesting in vitro work investigating responses to conditions in EColi that control anaerobic respiratory genes and antibiotic tolerance. The manuscript is well written and structured and easy for the reader to follow. The reviewer would like to bring a couple of points to the attention of the authors so that they take them into consideration. One important aspect in the discussion section is the potential approach and possible application in the context of the gut microbiome and/or the gut health and responses in the case of exposure. The reviewer would highly suggest the inclusion of a short paragraph in the discussion section discussing potential implication of the findings as well as a broader discussion on gut health and overall health implications in humans. A couple of papers that may be useful to that end for such a discussion are listed below:

1) Sikalidis AK, Maykish A (2020) The Gut Microbiome and Type 2 Diabetes Mellitus; discussing a complex relationship. Biomedicines. 8(1):8. doi.10.3390/biomedicines8010008.

2) Sikalidis AK (2015) Amino Acids and Immune Response: A role for cysteine, glutamine, phenylalanine, tryptophan and arginine in T-cell function and cancer? Pathol Oncol Res. 21(1):9-17. doi: 10.1007/s12253-014-9860-0.

Good work overall.

OUR ANSWER: As recommended, we included sentences mentioning the impact of the study for research of gut microbiome and gut health with citing the references the reviewer suggested (L30~31). We thought that referring it in the Introduction section is more suitable.

Round 2

Reviewer 3 Report

The authors have done well in addressing reviewer's comments and the manuscript has improved in the reviewer's professional opinion. I thus recommend acceptance for publication.